# The Challenging Management of Craniopharyngiomas in Adults: Time for a Reappraisal?

**DOI:** 10.3390/cancers14153831

**Published:** 2022-08-07

**Authors:** Thomas Cuny, Michael Buchfelder, Henry Dufour, Ashley Grossman, Blandine Gatta-Cherifi, Emmanuel Jouanneau, Gerald Raverot, Alexandre Vasiljevic, Frederic Castinetti

**Affiliations:** 1Aix Marseille University, MMG, INSERM U1251, MAEMARA Institute, Department of Endocrinology, CRMR HYPO, 13385 Marseille, France; 2Department of Neurosurgery, University Hospital of Erlangen, 91052 Erlangen, Germany; 3Aix Marseille University, MMG, INSERM U1251, MAEMARA Institute, Department of Neurosurgery, CRMR HYPO, 13385 Marseille, France; 4Centre for Endocrinology, Barts and the London School of Medicine, London OX1 2JD, UK; 5Department of Endocrinology, CHU of Bordeaux, 33000 Bordeaux, France; 6Neurocentre Magendie, University of Bordeaux, 33600 Pessac, France; 7Skull Base and Pituitary Neurosurgical Department, Reference Centre for Rare Pituitary Diseases HYPO, ‘Groupement Hospitalier Est’ Hospices Civils de Lyon, 69500 Bron, France; 8INSERM U1052, CNRS UMR5286, Claude Bernard Lyon 1 University, Cancer Research Center of Lyon, 69000 Lyon, France; 9Endocrinology Department, Reference Center for Rare Pituitary Diseases HYPO, ‘Groupement Hospitalier Est’ Hospices Civils de Lyon, 69500 Bron, France; 10East Center of Pathology and Neuropathology, “Groupement Hospitalier Est”, Hospices Civils de Lyon, 69500 Bron, France; 11Department of Endocrinology, La Conception Hospital, 147 Boulevard Baille, CEDEX 05, 13385 Marseille, France

**Keywords:** craniopharyngioma, adamantinomatous, papillary, targeted therapies, radiotherapy, neurosurgery

## Abstract

**Simple Summary:**

Craniopharyngiomas (CPs) currently represent one of the most challenging diseases to deal with in the group of skull base tumors. Due to their location near, within, or surrounding the pituitary gland and stalk, CPs can be revealed by pituitary tumor syndrome and/or symptoms of hormonal deficiencies. Furthermore, surgery, which represents the first-line therapy, almost always results in hypopituitarism, diabetes insipidus and, in the case of hypothalamic involvement by the tumor, the occurrence of hypothalamic syndrome. The latter is characterized by intractable weight gain associated with severe morbid obesity, memory impairment, attention deficit, reduced impulse control and, eventually, increased risk of cardiovascular and metabolic disorders. Recent progress made in the understanding of the molecular pathways involved in CPs tumorigenesis paves the way for promising alternative therapeutic approaches and diagnostic procedures. Taken together, they lay the groundwork for new paradigms in the management of CPs in adults.

**Abstract:**

Craniopharyngiomas (CPs) are rare tumors of the skull base, developing near the pituitary gland and hypothalamus and responsible for severe hormonal deficiencies and an overall increase in mortality rate. While surgery and radiotherapy represent the recommended first-line therapies for CPs, a new paradigm for treatment is currently emerging, as a consequence of accumulated knowledge concerning the molecular mechanisms involved in tumor growth, paving the way for anticipated use of targeted therapies. Significant clinical and basic research conducted in the field of CPs will undoubtedly constitute a real step forward for a better understanding of the behavior of these tumors and prevent associated complications. In this review, our aim is to summarize the multiple steps in the management of CPs in adults and emphasize the most recent studies that will contribute to advancing the diagnostic and therapeutic algorithms.

## 1. Introduction

Craniopharyngiomas (CPs) are rare brain tumors resulting from malformations of embryonic remnants along the original pathway of the craniopharyngeal duct [1]. Overall, CPs comprise 1.2 to 4.6% of all intracranial tumors, with an incidence of histologically confirmed cases of 0.16/100,000 persons per year in the USA [2]. In spite of belonging to the group of benign epithelial tumors, according to the World Health Organization [3], CPs are highly problematic in the clinical field because of the hormonal and hypothalamic disorders that they cause. In children, a recent observational study found a high prevalence of early endocrine disorders after brain tumors, including in children suffering from CPs [4]. Similar outcomes are seen in the adult population, with complications of visual, pituitary, and/or hypothalamic function, all of these regions being exposed to surgically induced damage that is associated with tumor resection. Moreover, cases of malignant CPs exist, even though their definition/presentation is not currently particularly well described [5,6,7]. CPs are divided into two distinct subtypes, adamantinomatous CP (ACP) and papillary CP (PCP), differing both in histological features and genetic alterations [8] (Figure 1). ACP is the most prevalent subtype seen in children and adults, displaying a bimodal age distribution, with peaks between the ages of 5–15 years and 45–60 years. Conversely, PCPs are almost exclusively encountered in adults. At the genetic level, somatic mutations in *CTNNB1* (encoding β-catenin) are found in roughly 90% of ACPs, leading to the activation of the WNT pathway, while PCPs frequently harbor somatic *BRAF_V600E_* mutations that result in the activation of the mitogen-activated protein kinase (MAPK) signaling pathway [9].

Symptoms at diagnosis in adults differ from those seen in children, with frequent visual alterations as presenting features of the patient because of the tumor mass effect. Headaches can be moderate; however, their chronic presentation makes them unusual and worrying for the patient. When present, endocrine deficiencies are responsible for individual impairments such as sexual dysfunction or polydipsia/polyuria. In parallel, patients may complain of disabilities affecting their social and professional lives, such as a decline in cognitive function with a significant impact on their job performance [10,11,12]. Albeit rare before surgery, a hypothalamic syndrome may be suspected preoperatively, when disruptions in body temperature regulation, growth, and water balance or eating behavior disorders are present. Even today, the prognosis in adult patients with CPs constitutes an important issue, as a 3-fold overall mortality rate and an up to 19-fold higher cerebrovascular mortality rate have been reported as compared to the general population [13,14,15]. Indeed, even when disease control is achieved, related disabilities frequently require the intervention of multiple health care professionals with, sometimes, an incapacity to return to normal professional activity. For CPs, one current challenging discussion concerns the treatment strategy which, for years, has relied almost exclusively on surgery and radiotherapy. Recent progress that has been made in deciphering the molecular background of CPs now paves the way for new therapeutic approaches.

## 2. Management of Craniopharyngiomas: A Multimodal Approach

Except for small localized craniopharyngiomas (CP) that can benefit from close monitoring, the classical management of CP most often involves a multimodal approach that may combine one or more surgical operations with various modalities of radiotherapy (including intra-cystic treatments). The following paragraphs briefly detail the main therapeutic modalities for CP, keeping in mind that these approaches should always be discussed in an expert center by a multidisciplinary team [16].

### 2.1. Surgery

Surgery can be proposed in several settings: as an emergency for chiasmatic decompression or reduction of intracranial pressure; as a preventive measure to avoid chiasmatic compression or intracranial hypertension in the case of recurrence; or to reduce the tumor volume which may then be subjected to radiotherapy [17]. The question of the degree of extension of the surgery has been controversial: some have advocated radical surgery, which could either lead to the removal of the pituitary stalk and/or damage to the hypothalamic structures, while others have proposed less radical surgery, with a higher risk of recurrence but less morbidity for the surrounding structures (especially the hypothalamus). The aim of treatment is to avoid or control recurrence, at the cost of a low risk of per- and postoperative morbidity. In particular, the risk of hypothalamic dysfunction induced by surgery, while sometimes already present at diagnosis, is one of the main criteria to consider [18]. The risk of complications induced by the degree of surgical resection can be assessed by preoperative MRI, which allows visualization of the degree of adhesion or compression of hypothalamic structures. Several classifications of hypothalamic morbidity correlated with post-operative abnormalities and the status of the third ventricle floor have been reported [19]. Of note, any surgery for tumor recurrence is associated with lower surgical efficacy than the original surgery, and an increased risk of complications, including mortality, which increasingly leads to a preference for second-stage radiotherapy instead of surgical revision. For pure cystic CPs, a cystic aperture in the third ventricle (sometimes with a sub-cutaneous reservoir) followed by a wait-and-see strategy may also represent a safe option [20].

### 2.2. Irradiation

Different modalities of radiation therapy have been evaluated in patients with CP. They include conventional external radiotherapy, proton beam therapy, stereotactic radiotherapy, and radiosurgery. Usually, these modalities are used in combination with partial surgery, or more rarely alone [21,22]. When combined with surgery, radiation techniques can be employed in two different settings: immediately after an incomplete surgery (partial surgery decided by the surgeon), or after surgery, during the follow-up of the patient, when a new tumor remnant appears, or if a previously known tumor remnant becomes progressive (radiotherapy as a second step). In that setting, the notion of slow progressiveness is important, since radiotherapy does not have an immediate effect on tumor volume, with a decrease in tumor volume generally starting from 6 to 12 months after the procedure and continuing over time. Historically, the most common modality of radiotherapy used in patients with CPs consisted of the delivery of a fractionated dose of photons (i.e., conventional external radiotherapy). With this approach, volume control of the tumor residue is high, estimated at nearly 80–90% by some studies. Another type of conformational therapy can be used with protons instead of photons (Proton therapy). It has the theoretical advantage of low proton scattering on the surrounding structures, thus reducing the risk of complications [23]. However, efficacy data for this approach remains rather limited to date. 

Techniques of radiation therapy using a stereotactic frame can be either used with fractionation (stereotactic radiotherapy) or with the delivery of a single fraction (also called radiosurgery). Interestingly, their precision is precious to spare adjacent structures. Radio-surgery modalities include Gamma Knife or cyber-knife. Data on pituitary tumors suggest a better tolerance but a lower efficacy of these more precise approaches compared to conventionally fractionated radiotherapy [24].

### 2.3. Intra-Cystic Treatments

Cysts are frequently found in the case of CPs and can therefore develop in the anatomical spaces nearby the pituitary sella. Amongst them, the formation of cysts occurring in the third ventricle is one of the most frequent localization of cystic development: it can lead to obstructive hydrocephalus either as the calling-in point at the initial diagnosis and/or in case of recurrence. It may require the use of an intra-cystic catheter or even an Ommaya reservoir to allow repeated evacuation of the cystic fluid. Regarding intra-cystic treatments, the data available in the literature is currently scarce. This approach only applies to craniopharyngiomas with exclusively, or predominantly, cystic recurrence. Intra-cystic treatment with gamma-interferon may be useful in about ¼ of cases, without a clear predictive factor of efficacy. A recent study demonstrated an absence of recurrence at 14 months of follow-up in 14 of 56 children treated with this approach [25]. Other molecules (bleomycin) or radioisotopes (^90^Yttrium and ^32^Phosphorus) have been proposed: in the absence of robust published data, the risk-benefit ratio does not seem generally to be in favor of this type of treatment [26]. However, a discussion may be held in a multidisciplinary consultation board to determine an individualized approach.

### 2.4. Side Effects and Quality of Life

#### 2.4.1. Side Effects of Treatments 

Side effects consist mainly of the occurrence of new hormonal deficiencies, in addition to those already present at diagnosis, the latter being mostly central hypogonadism and GH deficiency. Hypothalamic damage may also lead to obesity. Hormonal deficiencies justify lifelong hormone replacement therapy (pituitary deficiencies including AVP deficiency). The question of growth hormone (GH) replacement therapy and the possible risk of recurrence must be discussed on a case-by-case basis, as emphasized by recent recommendations concerning intracranial tumors [27]. Although in vitro data suggests a proliferative effect of GH in cell culture [28], clinical data are actually very reassuring and GH substitution appears to be beneficial for patients with GH deficiency [29,30,31]. The management of hypothalamic obesity is complex and has for a long time relied solely on strict dietary management. Growth hormones can also improve body composition in this setting. The potential benefits of medicinal (GLP-1 analogues, oxytocin) or surgical approaches (bariatric surgery) are currently being studied to evaluate their efficacy and their possible side effects [32,33,34].

Radiotherapy, especially if performed in the pediatric age group, is an effective treatment, but may expose the patient to later risk of radiation-induced tumors, cognitive disorders, or vascular thrombosis [35]. The indication for and modality of radiotherapy should therefore be determined by a multidisciplinary tumor board. Most retrospective studies evaluating the risks of radiotherapy have been conducted using older modalities in order to obtain data with a sufficiently prolonged follow-up. The most modern modalities (radiosurgery, stereotactic radiotherapy, and proton beam therapy) represent promising tools that deserve to be further evaluated in the specific context of craniopharyngiomas. Progress in the delivery of proton therapy could lead to a preference for this modality in the future because it could potentially reduce the risk of side effects; however, this remains to be demonstrated. 

#### 2.4.2. Quality of Life in Patients with Craniopharyngiomas

Patients with CP have an altered quality of life and often have cognitive or attentional problems [36,37]. However, there are specific data in the literature on the long-term quality of life of patients with CP, compared by therapeutic approach in terms of the different modalities of radiotherapy, or the approach of repeated surgery versus surgery combined with radiotherapy. It is likely that the degree of aggressiveness of its management (the number of different procedures, extension of surgery and associated post-operative morbidities) is correlated with the long-term quality of life, as was suggested in the short term (three-year follow-up) study from the KRANIOPARYNGEOM registry [38]. Psychological care is essential for all patients from the beginning of the treatment, and at each disease recurrence.

## 3. Molecular Landscape of Craniopharyngiomas

ACPs and PCPs are characterized by specific molecular signatures, each of the subtypes harboring alterations in oncogenic signaling pathways in a mutually exclusive way.

More than 70% of ACPs harbor a mutation of the *CTNNB1* gene, encoding the β-catenin protein [39,40,41]. Mutations in *CTNNB1* in ACP were first described in 2002 by Sekine et al. [40]. Recently, whole-exome sequencing as well as targeted genotyping found *CTNNB1* mutations in more than 90% of ACPs [9]. While other studies have reported a failure to detect *CTNNB1* mutations in a proportion of ACP samples, these results rather reflect methodological limitations such as an insensitive sequencing approach and/or a low proportion of the analyzed tumor tissue in the samples that were analyzed [42,43]. Activating mutations in *CTNNB1* found in ACP are i) clonal [43] and ii) mostly represented by point mutations within exon 3, which encodes the degradation targeting box of β-catenin. Consequently, the mutated protein is no longer degraded, resulting in its aberrant nucleo-cytoplasmic accumulation [40] and unsuppressed activation of the WNT/β-catenin pathway. Increased Wingless (Wnt) signaling has been shown to promote tumor transformation in pituitary progenitor/stem cells [44] through the transcription of target genes, such as CyclinD1, c-Myc, and CD44, involved in the process of cellular proliferation. Interestingly, no other mutations have been thus far identified as being a driver of ACP tumorigenesis, including genes that are found to be commonly mutated in other brain tumors [43]. Of clinical relevance, mutations in *CTNNB1* can be demonstrated by immunohistochemistry by showing nucleo-cytoplasmic accumulation of β-catenin either in single cells throughout the tumors or in small groups of cells referred to as cell clusters [45,46]. Usually, these cells showing β-catenin accumulation are observed either at the base of epithelial tumor protrusions towards the gliotic brain tissue, or in epithelial whorls (also referred to as clusters) and, importantly, have never been described in other types of brain tumors, including PCPs. Following these preliminary observations, a large body of research has been carried out using mouse models to better understand how *CTNNB1* mutations are involved in the pathogenesis of ACP [47]. The group headed by Martinez-Barbera showed in elegant studies that, when occurring in pituitary stem cells (namely SOX2^+^ cells), *CTNNB1* mutations led to the development of clusters of cells originating from these SOX2^+^ cells which, in turn, are able to secrete high levels of growth factors (epidermal growth factor, fibroblast growth factor, transforming growth factor beta and cytokines (such as sonic hedgehog, IL-1 and IL-6), that will eventually promote proliferation and invasion of the surrounding epithelial cells by the tumor [44,48]. These findings confer the tumor its unusual appearance of finger-like protrusions. Thus, this complex pro-tumorigenic molecular organization relies on paracrine regulation of tumor formation and growth, by the release of factors compatible with a pro-tumorigenic senescence-associated secretory phenotype by cells that are organized in clusters [49]. This senescent phenotype could be the cause (or the consequence) of shortened telomeres that have been observed in ACPs harboring mutations as compared to their counterparts that do not harbor mutations [50]. A recent study showed that, in addition to the mass effect exerted by the tumor, the hypothalamic sequelae of ACPs could also result from a direct and detrimental effect of inflammatory factors contained in the tumor cyst [51]. As such, by injecting human ACP cyst fluid into the bilateral hypothalamus of mice, growth retardation and increased obesity occurred together with a decreased expression of genes involved in growth regulation and energy metabolism in hypothalamic neurons. Eventually, β-amyloid deposition, a marker of neurodegenerative diseases, was also observed in the hypothalamic area of injected mice [51]. The immune system appears to also play a role in the pathogenesis of ACPs, as evidenced by the release of immunosuppressive factors, such as IL-10, in the ACP cyst fluid [48]. Moreover, program-death ligand 1 (PD-L1) and its receptors (PD-1), or in some cases other immune checkpoint proteins, for instance, belonging to the B7 family or VISTA proteins, have been found to be significantly expressed in ACPs [52,53]. Finally, an integrative genomic analysis recently suggested that particular types of immune cells, including M2 macrophages, activated NK cells, and gamma/delta T cells were overrepresented in ACPs as compared to CD8+ T cells, regulatory T cells, or neutrophils [54]. The ras-Raf-Mek-ERK1/2 signaling pathway has also been examined in the pathophysiology of ACP, as it is one of the most disrupted pathways in human cancers [55]. Interestingly, a previous study showed that the paracrine regulation of tumor cells by clusters in CPs involved the activation of the MAPKinase pathway, particularly at the forefront or leading edge of the tumor [48].

The MAPKinase pathway is also over-activated in PCPs but as a consequence of the *BRAF_V600E_* mutation. So far, no other molecular alterations have been identified in PCPs except for somatic *BRAF_V600E_* mutations [9]. The expression of oncogenic BRAF_V600E_ has been documented using targeted genotyping in 95% of PCPs and 100% of PCPs when using whole-exome sequencing [8,9]. BRAF belongs to the family of RAF serine/threonine kinases that have been studied extensively and have been shown to be implicated as oncogenes in many different cancers. The BRAF mutant acts as an oncogene as the mutant protein kinase domain remains in an open, active configuration which, in turn, activates downstream effectors like the extracellular signal-regulated kinase 1 and 2 (ERK1/2) [56]. Again, mutations in *BRAF_V600E_* can be reliably assessed through immunohistochemical methods, using anti-BRAF_V600E_ antibodies, showing a good correlation when compared with results obtained using Sanger sequencing in the same tumors [57]. Analysis of the tumor compartment in PCPs clearly showed that activation of the MAPK signaling pathway, secondary to *BRAF_V600E_* mutation, only occurred in a subset of cells, namely those that exhibit both a marker of precursor dedifferentiation (SOX2) and a proliferation marker (Ki-67). These cells are located specifically surrounding the fibrovascular core in both human and mouse models of PCP, and expression of BRAF_V600E_ in SOX2 pituitary progenitors blocks their normal further differentiation and promotes their proliferation [49]. As these cells are exclusively confined to the lining space of the fibrovascular core, the exact mechanisms by which growth of the tumor occurs remains elusive and, whether there is paracrine regulation, similar to that described in ACPs, merits further investigation. Overall, neither *CTNNB1*,nor *BRAF_V600E_* mutations have been shown to be associated with a more aggressive phenotype of CPs. However, another gene, claudin-1, a tight junction protein, has been shown to have a lower expression (at both mRNA and protein level) in CPs that displayed features of invasiveness, irrespective of CP subtype [58].

In summary, the molecular landscape of craniopharyngiomas is characterized by the existence of mutually exclusive *CTNNB1* and *BRAF_V600E_* mutations in ACPs and PCPs, respectively. However, a very small degree of overlap can also exist [57]. Because the vast majority of these tumors harbor mutations, presently, there is hope for the future development and use of targeted therapies. Currently, the main molecular pathways that would seem reasonable to target are those involved in activation of the WNT/β-catenin pathway, thus disrupting the paracrine regulation of tumor growth in ACPs, and/or the MAPkinase signaling pathway, whose activation has been observed in both histological subtypes. At the same time, new pathways may also be of interest, such as opposing or attenuating the inflammatory phenotype, a characteristic of CPs, using specific drugs, or supporting the immune response through the use of immune checkpoint inhibitors.

## 4. Future Directions for the Care of Craniopharyngiomas

### 4.1. Diagnosis

#### 4.1.1. Imaging

The diagnosis of a CP in adults is greatly suspected when there is a combination of specific features on brain magnetic resonance imaging (MRI), usually performed when clinical signs are suggestive of a pituitary tumor mass effect (i.e., visual field and/or acuity abnormalities, headaches) and/or hormonal deficiency. In the era of availability of modern imaging procedures, it should be emphasized that the rate of CPs diagnosed incidentally remains low, below 2% [59]. In this unusual situation, CPs are, as could be expected, asymptomatic, and characterized by a lack of endocrine deficiencies and a smaller tumor volume as compared to symptomatic CPs [59]. The diagnosis of CPs is made by using contemporary imaging including computerized tomography and MRI, which can identify suprasellar lesions with an intrasellar portion in the majority of cases, while only 20% are exclusively suprasellar and even fewer (5%) are exclusively intrasellar [60] (Figure 2). On an MRI, APCs typically present with three components: solid, cystic, and calcified parts which occupy the suprasellar cistern. The cystic component may be single or multiple and usually hyperintense on T1, T2 and FLAIR weighted images due to the presence of proteinaceous liquid. The solid component has variable signal intensities and usually shows contrast enhancement, while calcification is very common (90% of cases) and better demonstrated by T2 sequence imaging. Concerning PCPs, they usually appear as solid or mixed (predominantly solid and cystic) spherical tumors in the suprasellar region, with more upward growth towards the third ventricle. In general, they less frequently contain calcification; however, for both subtypes, there is a lack of specificity in the radiological appearance that can clearly differentiate ACPs from PCPs. Apart from patient age and the location of the tumor, recent concepts using artificial intelligence have been proposed for distinguishing ACPs from PCPs based on morphological imaging [61]. In 2018, Yue et al. showed in a series of 52 adult CPs patients that suprasellar (*p* < 0.001), spherical (*p* = 0.005), predominantly solid (*p* = 0.003), homogeneous enhancement (*p* < 0.001), and a thickened pituitary stalk (*p* = 0.014) were all significantly more frequently observed in tumors with *BRAF* mutations (i.e., PCPs) as compared to tumors without *BRAF* mutations (that were *de facto* mainly ACPs). It is noteworthy that, when at least three of these five features were present, the tumor was diagnosed as a PCP with a sensitivity and specificity of 100% and 91%, respectively [62]. Another study developed an MRI-based radiomics approach to differentiate ACPs (*n* = 22) from PCPs (*n* = 12) in a series of 44 patients, of whom 10 had non-mutated CP [63]. In this predictive model, the authors combined the use of high-throughput features regarding tumor location, intensity, shape, texture, and wavelet to differentiate between ACPs and PCPs based on pathology, with an AUC (area under the curve) of 0.89, accuracy of 0.86, sensitivity of 0.89 and specificity of 0.85 [63]. However, this approach still needs to be further validated on additional independent cohorts of patients. Another challenging aspect in the imaging of CPs is to predict the invasion of hypothalamic structures, which, ultimately, could guide neurosurgeons as to the best surgical strategy. In children, a radiological classification established by Puget et al. has been proposed to classify tumors into three grades, depending on the degree of hypothalamic involvement (grade zero = none, grade one = compression of the hypothalamus that can still be identified, and grade two when the hypothalamus is unidentifiable) [64]. Accordingly, gross total resection is possible in grades zero and one, while subtotal tumor resection should be actively discussed when the tumor is grade two, especially in adults [65]. A similar classification is not currently available in adults; however, a new preoperative MRI-based tool named the “eagle sign” has been recently retrospectively assessed in a series of 146 CP patients [66]. This radiological sign, bordered by the third ventricle floor, mamillary bodies and the cerebral peduncle on the scan, could predict hypothalamic infiltration, based on its orientation (namely upward versus downward), as observed during surgery, with a sensitivity and specificity of more than 90% respectively [66]. Previous work, conducted in a series of 500 CPs, demonstrated the accuracy of a model based on multiple variables which take into account the topography of the tumor, its consistency, and the presence of hydrocephalus or infundibulo-tuberal or hypothalamic syndrome at diagnosis, for predicting the severity of tumor adherence in 87% of cases [67].

#### 4.1.2. Liquid Biopsy

Recent progress made in the diagnosis of cancer includes the exponential development of the use of liquid biopsies [68]. In the field of CPs, no current study has assessed the feasibility of cerebrospinal fluid-based liquid biopsies to determine the mutational status of the tumor. Nonetheless, CPs, especially the papillary subtype, would represent candidates of choice for this type of molecular approach, as a response to BRAF inhibitors has been observed in early clinical trials (see below). A CSF test that could obviate a biopsy could be very helpful, as the risk of surgical intervention in these tumors is not insignificant [69]. Recent evidence has shown that the detection of cell-free DNA, including circulating tumoral DNA, is effective and even more accurate than classical cytological analysis, for the diagnosis of leptomeningeal disease from cerebrospinal fluid [70]. Whether a similar approach for the molecular diagnosis of CP subtypes is feasible is currently unknown.

### 4.2. Treatment

#### 4.2.1. Restricted Surgery and Radiotherapy

Although the primary treatment of CP relies on surgery and radiotherapy, recent advances in our knowledge of the molecular mechanisms involved in their initiation and growth considerably improve the perspective of disease control in CP patients. Risk-adapted surgical treatments in adults have been proposed to adapt the best surgical strategy, sparing the hypothalamus and protecting the patient from long-term sequelae such as obesity and disorders of temperature or thirst regulation [71]. As such, gross total resection of the tumor remains recommended when there is no infiltration of the hypothalamus, while it is recommended to perform a subtotal tumor resection followed by adjuvant radiotherapy when hypothalamic infiltration is confirmed [71]. Radiotherapy in adults with CPs results in an excellent control rate of 100% at the five-year follow-up, and 94% at the 10-year follow-up [72]. Visual deterioration and endocrine deficiencies following radiotherapy may occur in 10% to 20% of patients [72].

#### 4.2.2. Targeted Therapies

In the specific case of PCP, new therapeutic strategies have been discussed. There are several case reports in the literature that have shown an exceptional antitumoral response in a PCP harboring BRAFV600E mutation to the combination or single use of BRAF (Dabrafenib) and/or MEK (Trametinib) inhibitors [73,74,75,76,77,78,79,80,81], legitimately raising the question of using MAPK inhibitors as a neoadjuvant therapy before considering surgical debulking and/or radiotherapy. While off-label, a recent proof-of-concept study showed that performing a biopsy of the tumor via a trans-ventricular neuroendoscopic approach to obtain a sufficient tissue sample for histopathological and molecular analysis succeeded in (1) detecting a somatic BRAFV_600E_ mutation and (2) to drastically reducing tumor volume (90%) after combination therapy with BRAF and MEK inhibitors and (3) relieving patient symptoms [82]. Ultimately, this result prompted the authors to propose a tumor biopsy, via endoscopic transventricular/transsphenoidal approach, as soon as a CP is suspected and when complete resection of the tumor is not feasible. When considering ACPs, there are currently no medical therapies available that have shown similar results to those observed with BRAF/MEK inhibitors in PCPs. However, the recent demonstration that the MAPK/ERK pathway is activated in the finger-like protrusions of ACP, followed by reduced proliferation and increased apoptosis in explant cultures of human and mouse ACP treated with trametinib, indicates that new medical avenues may well emerge in the near future.

## 5. Conclusions and Unmet Needs

The management of CPs remains a challenge. Despite their non-malignant nature, CPs present a very high risk of recurrence, currently requiring a multimodal approach to avoid local complications at the cost of serious treatment-induced morbidities. The classical combined approach of surgery and radiation therapy, possibly repeated, is the first-line treatment for a CP. However, a distinction must be made between an emergency situation, in which a rapid increase in volume requires systematic surgery, and a monitoring situation, in which imaging highlights the slow evolution of a tumor remnant, and for which a radiotherapy modality, with the least toxicity possible, could be the preferred option. The difference between radical or selective surgery in the first line should be the subject of large-scale multicenter studies, taking into account the risk of recurrence in comparison with the patient outcome in terms of hypothalamic–pituitary sequelae and quality of life.

However, advances in the understanding of the mechanisms underlying PCs could drastically change their management in the future: extensive surgery could be replaced by minimal surgery, indicated only in cases of an urgent threat (e.g., visual), or even by biopsy, which would allow the identification of a somatic genetic abnormality and the proposal of individualized management with targeted therapy (Figure 3). A first-line drug approach would probably avoid some of the hypothalamic and cognitive sequelae induced by surgery. However, we are only at the beginning of this paradigm shift, which will require prolonged follow-up studies to determine its potential merits and pitfalls (efficacy, escape, tolerance, etc.).

While we await these possible changes, the medical community must focus on improving the management of sequelae and, in particular the hypothalamic syndrome, a situation in which strict dietary management often has limited effectiveness. Data concerning drug treatments with an effect on satiety, or bariatric surgery are limited and often contradictory, justifying new prospective multicenter studies. Growth hormone treatment, as part of hormonal replacement therapy, can probably improve the patient quality of life and is essential in children who have growth delays. The management of cognitive and attention disorders requires regular psychological and neuropsychological follow-up from the time of diagnosis, and within the framework of therapeutic education for these complex patients, especially since recurrence often leads to an aggravation of pre-existing morbidities. Moreover, preventing or, at least, minimizing the sequelae of any management of CPs is essential. It requires the intervention of multidisciplinary team in an expert center, as well as national and international collaborations to propose innovative management strategies.

## Figures and Tables

**Figure 1 cancers-14-03831-f001:**
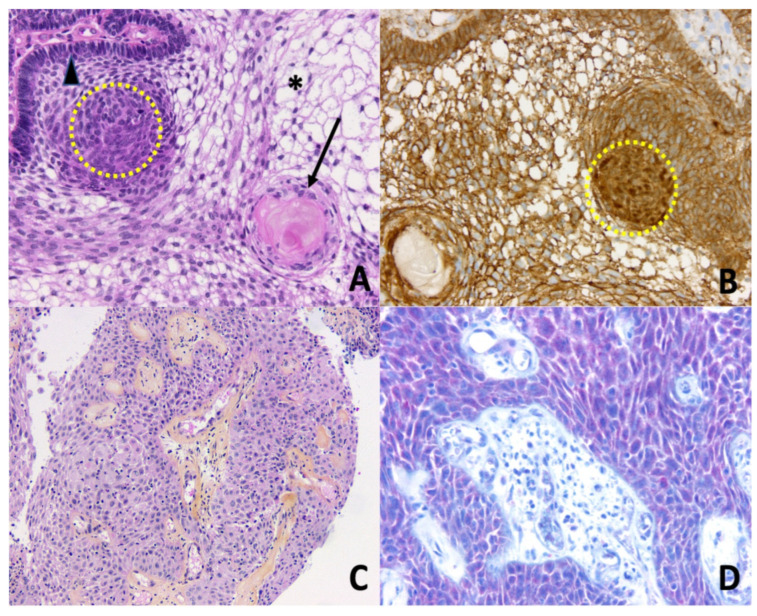
Histopathology of craniopharyngiomas. (**A**) Adamantinomatous craniopharyngioma. Epithelial nests with peripheral palisading columnar epithelium (arrowhead), nodular whorls (yellow dotted circle), stellate reticulum (asterisk), and aggregates of ‘wet’ keratin (black arrow) (Hematoxylin Phloxine Saffron (HPS) staining, ×200); (**B**) Adamantinomatous craniopharyngioma. Nucleocytoplasmic translocation and accumulation of β-catenin are detected by immunohistochemistry, especially in the nodular whorls (brown nuclear immunopositivity, yellow dotted circle, ×200); (**C**) Papillary craniopharyngioma. Papillae are composed of fibrovascular cores covered by a well-differentiated non-keratinizing squamous epithelium (HPS, ×100); (**D**) Papillary craniopharyngioma. Detection of BRAF V600E mutation by immunohistochemistry is positive (red cytoplasmic immunopositivity in squamous neoplastic cells, ×200).

**Figure 2 cancers-14-03831-f002:**
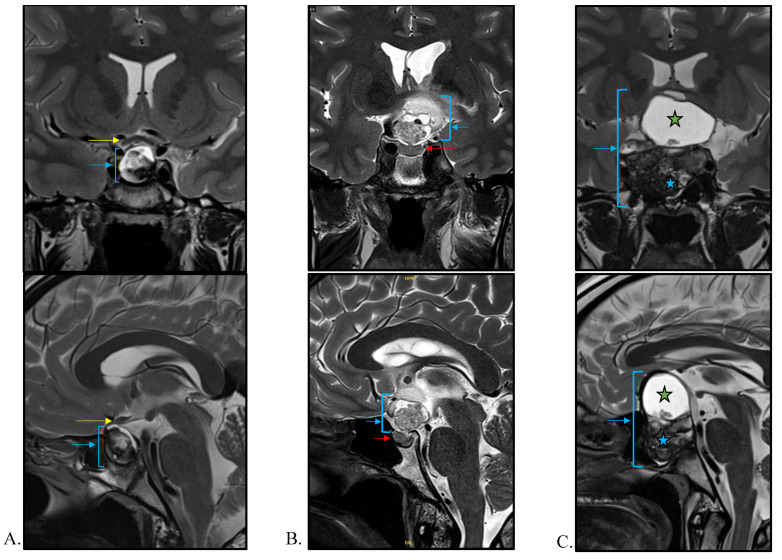
Neuroradiological characteristics of craniopharyngiomas (CP) in T2-weighted coronal and sagittal MRI. (**A**) Intrasellar CP (blue arrow) with a moderate suprasellar extension outcropping the optic chiasm (yellow arrow). (**B**) Exclusively suprasellar CP (blue arrow) showing heterogeneous content and respect to the pituitary gland (red arrow). (**C**) Voluminous CP with invasion of sellar and suprasellar regions (blue arrow). The pituitary gland, stalk, and optic chiasm are no longer visible. The upper part of the tumor is cystic (T2-hyperintensity, green star) while the bottom is fleshy and heterogeneous (blue star).

**Figure 3 cancers-14-03831-f003:**
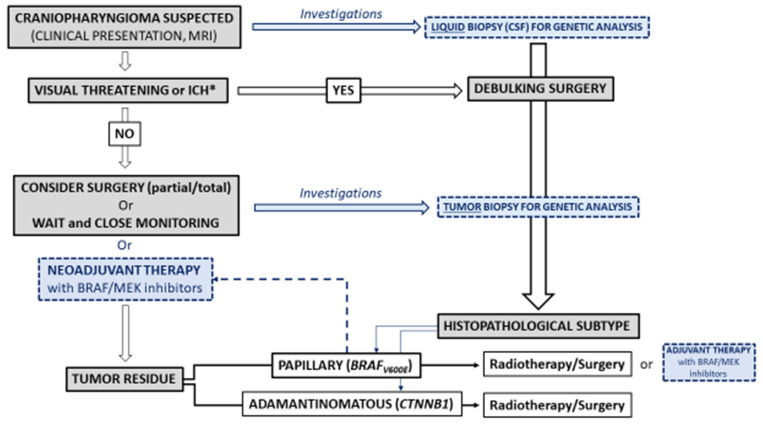
Schematic representation of the current and the new paradigm in the management of craniopharyngiomas (CPs) in adults. In blue and dotted lines, current or incoming investigations in the field of CPs* Intracranial Hypertension.

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
