# Peer review of "The Challenging Management of Craniopharyngiomas in Adults: Time for a Reappraisal?"

_cancers, 2022, doi:10.3390/cancers14153831_

Round 1

Reviewer 1 Report

This review proposes to reevaluate the management of craniopharyngiomas in adult patients by including targeted medical treatment based on molecular defined characteristics to a multidisciplinary approach based on advances in imaging, radiotherapy and on long-time follow-up studies of treatment-related morbidity.

General comments:

The topic  being  relevant to the field, the strenght within this review being the comprehensive presentation of the current state in understanding the molecular landscape as it relates to histopathological variants - with several reviews of these aspects having been published recently and referenced by the authors.

Integrating a targeted drug into the management of the small subgroup of papillary CPs could certainly play a role and has alreasdy been published as a feasable concept  by the Lyon group of coauthors of this review.

Weak points of this review that would definitely need to be revised substantially are the discussion (and headings) of radiation therapy, treatment options of cysts, as well as of Part 4. Regarding the role of (differnt types of) surgical interventions the current consensus on the surgical management of CPs in adult as published by the EANS skull base section (and referred to by the authors of this review) that includes relevant expert opinions on pre (-and post-) surgical as well as general management decisions could be pointed at more explicitely in my opinion.

Specific comments:

1) row 95-96, row 138, 396: why "recurrent tumors"? This is very confusing.

2) row 104-106: not preserving the stalk and damaging the hypothalamus are not strictly related. Restate or clarify.

3.) row 121: what is meant by "approach to recurrence" ?(see comment 1). Or do the authors mean "to reduce recurrence rate"?

4.) row 85, row 363: using the word "cure" seems inappropriate to me. Disease control would be the correct term.

5.) rows 119-136: Why "Radiotherapy and other options" when only radiotherapy is discussed? Also the presentation of radiotherapeutic options is not clear: radiosurgery (Gamma-knife, Cyberknife) and fractionated radiotherapy (be it with photons or protons, usually with stereotactic planning....

6.) row 138: apart from the unclear reference to recurrence (see comment 1), why only refer to cysts in the 3rd ventricle?

7.) row 179: delete "is"

8.) row 269: change line, this information refers to both subgroups.

9.) paragraph 4.1.: including the still only theoretical potential of liquid biopsy (for CP!) into the same discussion point as the proven and improving relevance of advanced imaging seems confusing.

10.) row 361-390:  a subheading of this paragraph seems to be missing.

11.) Several points in the Conclusions should be reformulated, specifically:

    a.) "local compressive complications at the cost of serious comorbidities": not only compression, are these complications ?, comorbidities OR morbidities (of the treatment)?

    b.) radiotherapy in the setting described (row 398) "could" but not necessarely "would" the preferred modality.

    c.) row 417-418: as the review is adressing the whole multidisciplinary team the imperative is not only to "improve" but most important to prevent or at least minimize the sequelae of any management.

Author Response

Reviewer#1

This review proposes to reevaluate the management of craniopharyngiomas in adult patients by including targeted medical treatment based on molecular defined characteristics to a multidisciplinary approach based on advances in imaging, radiotherapy and on long-time follow-up studies of treatment-related morbidity.

 General comments:

The topic being  relevant to the field, the strenght within this review being the comprehensive presentation of the current state in understanding the molecular landscape as it relates to histopathological variants - with several reviews of these aspects having been published recently and referenced by the authors.

Integrating a targeted drug into the management of the small subgroup of papillary CPs could certainly play a role and has already been published as a feasable concept  by the Lyon group of coauthors of this review.

Weak points of this review that would definitely need to be revised substantially are the discussion (and headings) of radiation therapy, treatment options of cysts, as well as of Part 4. Regarding the role of (differnt types of) surgical interventions the current consensus on the surgical management of CPs in adult as published by the EANS skull base section (and referred to by the authors of this review) that includes relevant expert opinions on pre (-and post-) surgical as well as general management decisions could be pointed at more explicitely in my opinion.

Answer: we thank the Reviewer#1 for his constructive criticisms. Accordingly, we substantially revised the points mentioned by the reviewer to improve the clarity of the manuscript

Specific comments:

1) row 95-96, row 138, 396: why "recurrent tumors"? This is very confusing.

Answer: we thank the Reviewer#1 for this relevant remark. We have, now, deleted the qualification “recurrent” in the revised version of the manuscript.

2) row 104-106: not preserving the stalk and damaging the hypothalamus are not strictly related. Restate or clarify.

Answer: we thank the Reviewer#1 for this important point. Accordingly, we modified the corresponding sentence in the revision (p. 3, l.113-118)

3.) row 121: what is meant by "approach to recurrence" ?(see comment 1). Or do the authors mean "to reduce recurrence rate"?

Answer: we thank the Reviewer#1 for this point that, indeed, needed a clarification. The sentence has been rephrased accordingly (p. 4, l.132-141)

4.) row 85, row 363: using the word "cure" seems inappropriate to me. Disease control would be the correct term.

Answer: we thank the Reviewer#1 for this suggestion, that, is used in the revised version of the manuscript now (p. 3, l.96 and p. 9, l.401)

5.) rows 119-136: Why "Radiotherapy and other options" when only radiotherapy is discussed? Also the presentation of radiotherapeutic options is not clear: radiosurgery (Gamma-knife, Cyberknife) and fractionated radiotherapy (be it with photons or protons, usually with stereotactic planning....

Answer: we thank the Reviewer#1 for this important point. We have modified the section 2.2 of the manuscript and its headings (p. 4, l.132 –164)

6.) row 138: apart from the unclear reference to recurrence (see comment 1), why only refer to cysts in the 3rd ventricle?

Answer: we thank the Reviewer#1 for this underlining this point. However, the aim of this review was not to detail the different of type of cysts encountered in the setting of CPs. As such, we clarified in the revised version that cyst developed in the 3rd ventricle are the most common seen in CPs (p.4, l.167-172)

7.) row 179: delete "is"

Answer: we thank the Reviewer#1 for this typo. We deleted it.

8.) row 269: change line, this information refers to both subgroups.

Answer: The line has been changed accordingly (p.7, l.280)

9.) paragraph 4.1.: including the still only theoretical potential of liquid biopsy (for CP!) into the same discussion point as the proven and improving relevance of advanced imaging seems confusing.

Answer: we thank the Reviewer#1 for this relevant remark on the form of the paragraph. We have, now, distinguished 2 subsections in the paragraph “Diagnosis” to avoid confusion.

10.) row 361-390:  a subheading of this paragraph seems to be missing.

Answer: we thank the Reviewer#1 for this remark. Accordingly, we distinguished 2 subsections in the paragraph 4.2 Treatment to separate surgery/radiotherapy from targeted therapies.

11.) Several points in the Conclusions should be reformulated, specifically:

    a.) "local compressive complications at the cost of serious comorbidities": not only compression, are these complications ?, comorbidities OR morbidities (of the treatment)?

Answer: we thank the Reviewer#1 for stressing out this sentence. We rephrased it in the revised version of the manuscript (p.10, l.437-439)

    b.) radiotherapy in the setting described (row 398) "could" but not necessarely "would" the preferred modality.

Answer: we thank the Reviewer#1 for this suggestion. We replaced “would” by “could” in the revised version of the text (p. 10, l.444)

    c.) row 417-418: as the review is adressing the whole multidisciplinary team the imperative is not only to "improve" but most important to prevent or at least minimize the sequelae of any management.

Answer: we thank the Reviewer#1 for this important remark. We rephrased the sentence in the revised version of the manuscript (p.11, l.471-475)

Reviewer 2 Report

Craniopharyngeomas in adults are a rare but challenging disease. The authors provide an excellent review on standard of care surgery and radiation as well as focus on novel therapeutic approaches such as BRAF/MEK inhibitors. In the conclusion the authors provide a treatment algorithm based on the current literature to provide guidance for the daily clinical practice.

Author Response

Reviewer#2

Craniopharyngeomas in adults are a rare but challenging disease. The authors provide an excellent review on standard of care surgery and radiation as well as focus on novel therapeutic approaches such as BRAF/MEK inhibitors. In the conclusion the authors provide a treatment algorithm based on the current literature to provide guidance for the daily clinical practice.

Answer: we thank the Reviewer#2 for his positive comments.

Reviewer 3 Report

The Authors present a review of the literature aimed at updating very rare tumors as craniopharyngiomas (CPs). In spite of belonging to the group of benign epithelial tumors, according the 2021 WHO classification of tumors of the central nervous system, CPs are burdened by a high percentage of endocrinological sequelae associated in some cases with decline in cognitive function. The current treatment strategy includes surgery and radiotherapy, but a high risk of relapse are reported. Improving knowledge of the biological characteristics of the tumor can improve the therapeutic approach. 

The proposed review topic is interesting and relevant both to modify the therapeutic approach and the management of sequelae; the bibliographic reference are appropriated; the conclusion drawn are consistent and supported. Although it is a narrative review, it provides an advancement of current knowledge, is interesting and could represent an important suggestion for cooperative studies.

Author Response

Reviewer#3

The Authors present a review of the literature aimed at updating very rare tumors as craniopharyngiomas (CPs). In spite of belonging to the group of benign epithelial tumors, according the 2021 WHO classification of tumors of the central nervous system, CPs are burdened by a high percentage of endocrinological sequelae associated in some cases with decline in cognitive function. The current treatment strategy includes surgery and radiotherapy, but a high risk of relapse are reported. Improving knowledge of the biological characteristics of the tumor can improve the therapeutic approach. 

The proposed review topic is interesting and relevant both to modify the therapeutic approach and the management of sequelae; the bibliographic reference are appropriated; the conclusion drawn are consistent and supported. Although it is a narrative review, it provides an advancement of current knowledge, is interesting and could represent an important suggestion for cooperative studies.

Answer: we thank the Reviewer#3 for his encouraging and positive comments.

Round 2

Reviewer 1 Report

In this revised form the paper is a welcome contribution to support cooperative efforts at improving the management of these difficult tumors by also including recent advances in medical treatment options.